# Structure and Characterization of Phosphoglucomutase 5 from Atlantic and Baltic Herring—An Inactive Enzyme with Intact Substrate Binding

**DOI:** 10.3390/biom10121631

**Published:** 2020-12-03

**Authors:** Robert Gustafsson, Ulrich Eckhard, Weihua Ye, Erik D. Enbody, Mats Pettersson, Per Jemth, Leif Andersson, Maria Selmer

**Affiliations:** 1Department of Cell and Molecular Biology, Uppsala University, BMC, Box 596, SE-751 24 Uppsala, Sweden; robert.gustafsson@icm.uu.se (R.G.); ueccri@ibmb.csic.es (U.E.); 2Department of Medical Biochemistry and Microbiology, Uppsala University, BMC, Box 582, SE-751 23 Uppsala, Sweden; weihua.ye@imbim.uu.se (W.Y.); erik.enbody@imbim.uu.se (E.D.E.); mats.pettersson@imbim.uu.se (M.P.); per.jemth@imbim.uu.se (P.J.); Leif.Andersson@imbim.uu.se (L.A.); 3Department of Veterinary Integrative Biosciences, Texas A & M University, College Station, TX 77843, USA; 4Department of Animal Breeding and Genetics, Swedish University of Agricultural Sciences, SE-75007 Uppsala, Sweden

**Keywords:** phosphoglucomutase 5, herring, adaptation, enzyme structure, phosphohexomutase

## Abstract

Phosphoglucomutase 5 (PGM5) in humans is known as a structural muscle protein without enzymatic activity, but detailed understanding of its function is lacking. PGM5 belongs to the alpha-D-phosphohexomutase family and is closely related to the enzymatically active metabolic enzyme PGM1. In the Atlantic herring, *Clupea harengus*, *PGM5* is one of the genes strongly associated with ecological adaptation to the brackish Baltic Sea. We here present the first crystal structures of PGM5, from the Atlantic and Baltic herring, differing by a single substitution Ala330Val. The structure of PGM5 is overall highly similar to structures of PGM1. The structure of the Baltic herring PGM5 in complex with the substrate glucose-1-phosphate shows conserved substrate binding and active site compared to human PGM1, but both PGM5 variants lack phosphoglucomutase activity under the tested conditions. Structure comparison and sequence analysis of PGM5 and PGM1 from fish and mammals suggest that the lacking enzymatic activity of PGM5 is related to differences in active-site loops that are important for flipping of the reaction intermediate. The Ala330Val substitution does not alter structure or biophysical properties of PGM5 but, due to its surface-exposed location, could affect interactions with protein-binding partners.

## 1. Introduction

The Baltic Sea is a unique environment. The salinity of the brackish waters gradually drops to about 2–3‰ in the Bothnian Bay compared to 35‰ in the Atlantic Ocean [1]. Herring is one of few marine fishes that reproduce throughout the Baltic Sea. Already Linneaus divided the Atlantic and Baltic herring into two separate subspecies, *Clupea harengus harengus* and *Clupea harengus membras* [2], but early genetic studies based on a limited number of markers were unable to separate them genetically [3]. Thus, it was thought that the difference in appearance between the small and lean Baltic herring and the larger and fatter Atlantic herring was purely from environmental origin. More recent studies based on whole genome sequencing of population samples have completely changed the picture and revealed hundreds of loci showing strong genetic differentiation between the Atlantic and Baltic herring [1,4,5]. This ecological adaptation must be recent, as the Baltic Sea has only existed since the last glaciation about 10,000 years ago [6]. The allele frequencies at several of these loci show a strong correlation with salinity at the spawning site where the fish were collected. An important discovery in these studies was that missense mutations have played a prominent role for the adaptation of herring to the brackish Baltic Sea [1]. For instance, it was recently reported that a missense mutation in rhodopsin (Phe261Tyr) has been critical for the adaptation to the red-shifted light environment in the Baltic Sea [7]. Here, we explore the functional significance of Ala330Val in phosphoglucomutase 5 (PGM5), another missense mutation that shows a striking allele frequency difference (>80%) between the Atlantic and Baltic herring.

PGM5 belongs to the alpha-D-phosphohexomutase family that most notably includes PGM1, PGM2, PGM2L1, PGM3, and PGM5 [8,9,10,11]. PGM5 is most closely related to PGM1 [8,9,10], which is the most studied PGM and the main isomerase responsible for catalyzing the interconversion between glucose-1-phosphate (G1P) and glucose-6-phosphate (G6P) (Figure 1) [9]. PGM1 is thus responsible for the metabolic shift between glycolysis (which uses G6P) and gluconeogenesis (which uses G1P) [12]. This reaction requires a phosphorylated serine residue that donates its phosphoryl group to the substrate to form the intermediate glucose-1,6-bisphosphate (G16P), which has to flip 180 degrees to allow transfer of the other phosphoryl group to the enzyme to regenerate the phosphoserine (Figure 1) [9]. The phosphoryl transfer reaction is reversible and is believed to occur using general acid-base catalysis [13]. Based on mutagenesis studies in bacteria [14,15,16], a lysine in a loop of domain III was proposed as catalytic base and an arginine in a loop from domain I as catalytic acid [12]. However, based on quantum mechanical calculations, an alternative catalytic acid histidine following the phosphoserine in loop 1 was recently proposed, suggesting that the arginine instead plays a role in electrostatic stabilization [17].

There is until now no structure available of PGM5, but several crystal structures of apo and ligand-bound PGM1 from different organisms, including rabbit and human [10,12,18,19]. These PGM1 structures are well-conserved and are composed of four domains (I–IV) that form a bi-lobed heart-shaped structure with a deep and wide active-site cleft between the first two and the last two domains. Upon ligand binding, domain IV closes the active site by a hinge-like movement creating a closed conformation [20,21,22]. One loop from each of the four domains, named 1–4 as the corresponding domains, contributes to the active site. Loop 1 includes the catalytic phosphoserine, loop 2 coordinates a positively charged divalent metal ion, loop 3 binds to the sugar, and lastly, loop 4 binds to the phosphate group of the ligand.

Human PGM5 was first isolated from muscle cells and called aciculin [23,24]. It was found to locate at adherens-type cellular junctions of muscle and some non-muscle cells and to be similar in sequence to the PGM family and human PGM1 [23,25]. However, human PGM5 was quickly found to lack phosphoglucomutase activity [24]. Instead, a non-enzymatic role of human PGM5 was suggested, since it was found to associate with the large cytoskeletal proteins dystrophin and utrophin [26,27]. Recently, human PGM5 was also found to interact with Filamin C and Xin, multi-adaptor proteins expressed in cardiac and skeletal muscles and important in the assembly and repair of myofibrils and their attachment to the membrane [28,29]. Filamin C binds to the N-terminus of PGM5 [28,29], while XinA and XinB bind to the C-terminal third of PGM5 [28]. Human PGM5 is thus in some manner involved in myofibril formation, maintenance, and remodeling [28]. The similar importance of PGM5 in fish was demonstrated in a PGM5 knockdown zebrafish where the embryos developed severe cardiac and skeletal myopathy and showed indications of paralysis [28]. Downregulation of human PGM5 has recently been identified as a prominent biomarker in several types of cancer [30,31,32,33]. In addition, the mutation Ile98Val is found to be oncogenic in stomach cancer [34,35]. In addition, a long non-coding anti-sense RNA from the *PGM5* gene has been shown to be implicated in colorectal cancer and brain tumors [30,36,37].

We here present crystal structures of herring PGM5, the first structures of this protein from any species. Furthermore, we have solved the structure of the most common PGM5 variants found in the Baltic and Atlantic herring (bPGM5 and aPGM5), differing by only one amino acid substitution (Ala330Val), and explore the possible functional significance of the mutation. We also describe a structure in complex with the substrate G1P and use a combination of structure and sequence comparison to shed light on why PGM5, despite its high similarity to PGM1, lacks PGM activity. Lastly, we identify intriguing differences between PGM5 from fish and mammals.

## 2. Materials and Methods

### 2.1. Genetic Analysis of PGM5 in the Atlantic Herring

The gene architecture of the Atlantic herring was explored using the most recent assembly of the herring genome (Pettersson et al., 2019). The analysis of delta allele frequencies between the Atlantic and Baltic herring as well as the allele frequency distribution in population samples from the Atlantic Ocean and the Baltic Sea was based on data reported by Han et al. (2020) [5]. The list of localities for herring samples included in this study is found in Appendix A. Sequences of transcripts for PGM5 predicted from ENSEMBL-provided annotation of the herring genome is found in Appendix A.

### 2.2. Molecular Cloning

The *PGM5* sequences from *Clupea harengus harengus* (Atlantic variant with Ala330) and *Clupea harengus membras* (Baltic variant: NCBI RefSeq XM_012820641.2) were codon-optimized for *E. coli*, synthesized with flanking restriction sites (NcoI and BamHI), and cloned into the pNIC28-Bsa4 plasmid encoding an N-terminal TEV-cleavable His6-tag (GENEWIZ UK, Takeley, UK). Resulting protein sequences are found in Appendix A. Plasmids were verified using DNA sequencing.

### 2.3. Protein Expression and Purification

Protein expression was done in *E. coli* BL21 (DE3) cells and single colonies used to inoculate overnight cultures in LB medium supplemented with 30 µg/mL kanamycin. Saturated cultures were diluted 1:1000 in fresh LB-medium containing kanamycin, incubated at 37 °C until OD(600) of 1.5–1.8 and induced with 1.0 mM isopropyl-β-D-thiogalactoside, followed by overnight incubation at 16 °C.

Cells were harvested by centrifugation for 20 min at 6000 RPM in a JLA 9.1000 rotor (5300 RCF, Beckman Coulter Inc, Brea, CA, USA) at 4 °C, and pellets resuspended in Buffer A (50 mM NaH_2_PO_4_, 300 mM NaCl, 10 mM imidazole, pH 8.0) including DNase I (10 µg/mL) and 10 mM MgCl_2_. The cell suspensions were lysed using a continuous flow cell disruptor (Constant Systems LTD, Daventry, UK) at 35 kPsi. Cell debris was removed by two-times centrifugation for ≥20 min at 15,000 RPM in an SS-34 rotor (26800 RCF, Thermo, Osterode, Germany) at 4 °C.

The clarified cell lysate was filtered through a 0.45 μm syringe filter, loaded onto a gravity column containing Ni-Sepharose 6 Fast Flow resin (GE Healthcare, Uppsala, Sweden) equilibrated in buffer A and incubated under slow rotation for ≥30 min at 8 °C. The column was washed using 10 column volumes (i.e., 10 mL of buffer per mL of resin) of (i) Buffer A, (ii) Buffer A with 1.0 M NaCl, (iii) Buffer A with 10 mM NaCl, and then (iv), Buffer A with 20 mM imidazole. The His6-tagged target protein was eluted using Buffer A supplemented with 250 mM imidazole and concentrated using ultrafiltration (Vivaspin Turbo 15, Sartorius; MWCO 10 kDa). The sample was loaded onto a Hiload 16/60 Superdex 200 column (GE Healthcare, Uppsala, Sweden) equilibrated with Buffer B (25 mM Tris-HCl, 150 mM NaCl, pH 8.0), after which the peak fractions were pooled and incubated overnight with 1:10 ratio of tobacco etch virus (TEV) protease [38] to protein in presence of 5 mM beta-mercaptoethanol at 8 °C. Non-cleaved target protein and the His6-tagged TEV protease were removed in Ni Sepharose re-chromatography. The flow-through containing tag-free target protein was concentrated and loaded onto a Hiload 16/60 Superdex 75 column (GE Healthcare, Uppsala, Sweden) equilibrated in Buffer B. For purification of His6-tagged protein, the peak fractions from HiLoad 16/60 Superdex 200 column were instead directly concentrated to 4 mL and loaded onto a HiLoad 16/60 Superdex 75 column equilibrated in Buffer B. Peak fractions were concentrated (Vivaspin Turbo 15, Sartorius; MWCO 10 kDa) and flash-frozen in liquid nitrogen. Of note, we did not quantitate the phosphorylation level of the active site serine in our protein preparations. Thus, we cannot exclude batch to batch variations, which may impact thermal stability due to effects on structural flexibility [39].

### 2.4. Protein Crystallization and Structure Determination

Crystallization was performed using the sitting-drop vapor diffusion method at room temperature using a mosquito crystallization robot (TPP Labtech, Melbourn, UK). Initial crystals were obtained for both aPGM5 and bPGM5 at 10–15 mg/mL concentration in the JCSG+ screen (Molecular Dimensions, Sheffield, UK). Conditions were subsequently refined to 1.0 M sodium malonate pH 6.0, 0.1 M MES pH 6.0, 1.0% (*v*/*v*) Jeffamine ED-2001 for His6-tagged aPGM5 and 0.12–0.16 M calcium acetate, 0.08 M sodium cacodylate pH 6.5, 14.4–15.0% (*v*/*v*) PEG 8000, 20% (*v*/*v*) glycerol for His6-tagged bPGM5. Crystals grew in 5–10 days and were directly vitrified in liquid nitrogen for data collection at Diamond Light Source (DLS, Didcot, UK) beamline I04 at wavelength 0.9795 Å using Pilatus 6M detector (Dectris) for aPGM5 apo crystals or ESRF (Grenoble, France) beamline ID29 at wavelength 1.07227 Å using Pilatus 6M-F detector (Dectris) for bPGM5 crystals. A complex structure of bPGM5 with G1P was obtained after overnight soaking of bPGM5 apo crystals with 5 mM alpha-D-Glucose-1,6-bisphosphate (Cayman Chemical, Ann Arbor, MI, USA, #16464). Diffraction data were processed using MOSFLM [40] and Aimless [41], and the structures solved using molecular replacement using either MoRDa [42] (aPGM5 apo, search model PGM1 structure 5EPC), Molrep [42] (bPGM5 + G1P, search model aPGM5 apo), or Phaser [43] (bPGM5 apo, search model bPGM5 + G1P). Structures were iteratively refined and manually rebuilt using phenix.refine [44] and Coot [45], within the macromolecular structure determination software suite Phenix [46]. Data and refinement statistics are summarized in Table 1. The structures have been deposited in the Protein Data Bank (PDB) with accession codes 6Y8X (aPGM5 apo), 6Y8Y (bPGM5 + G1P), and 6Y8Z (bPGM5 apo).

### 2.5. Enzymatic Assays

Phosphoglucomutase activity was measured using the PicoProbe Fluorometric Phosphoglucomutase Assay Kit (BioVision, San Francisco Bay area, CA, USA, #K770-100). Samples were prepared according to manufacturer’s instructions with 3.5 μM tag-free protein per well in 50 μL final reaction volume in half-area 96-well microplates (Corning, Fisher Scientific, Loughborough, UK). For each protein, reaction mixes were prepared in triplicates without additive, in the presence of 10 mM MgCl_2_, 1 μM G16P, or both. Measurements were performed in a Tecan Spark 10M microplate reader with 535/587 nm excitation/emission (20 and 20 nm bandwidth, respectively), recording the fluorescence signal in bottom reading setting every 60 s during 1 h at 26 °C. Samples were mixed prior to measurements by orbital shaking for 4 s.

### 2.6. Differential Scanning Fluorimetry

Thermal stability of tag-free aPGM5 and bPGM5 was evaluated by differential scanning fluorimetry following a protocol adapted from [47,48]. Each 25 µL reaction consisted of 5 µM untagged protein, 25 mM Tris-HCl, 150 mM NaCl, pH 7.5, and 5x SYPRO orange dye. SYPRO orange dye with buffer served as negative control. For experiments in the presence of G16P, proteins were preincubated with 500 µM ligand for 15 min at 4 °C before the addition of dye. Assays were performed in triplicates in a Bio-Rad CFX Connect Real-Time PCR Detection System (Bio-Rad Laboratories, Hercules, CA, USA), and samples were heated from 15 to 95 °C at 0.5 °C per minute, with measurement every 0.2 °C. The fluorescence intensity was measured and the melting temperature (T_m_) determined with the Bio-RAD CFX Manager Program and NAMI [49].

### 2.7. Circular Dichroism Spectroscopy

Circular dichroism (CD) measurements were performed on a Jasco J-1500 CD spectrometer (Jasco, Tokyo, Japan) equipped with an MCB-100 circulation bath as temperature controller. Quartz cuvettes with a light path of 1 mm were used for all measurements. CD spectra were recorded between 196 and 260 nm with a step of 0.1 nm at 25 °C. A bandwidth of 1 nm and a scanning speed of 50 nm/min in a continuous mode were used. Spectra were collected and averaged over 4 accumulations. The thermal denaturation experiments were monitored by the CD signal at 222 nm in the range of 4 to 95 °C with a step of 0.2 or 1 °C and temperature ramping rate of 1 °C/min. Protein with a concentration in the range of 1–8 μM in 50 mM sodium phosphate buffer (pH 7.4) was used. Both His6-tagged protein and tag-free protein were used for CD measurements for aPGM5 and bPGM5. The representative data in Appendix A are for tag-free protein, and the representative data in Appendix A are for His6-tagged protein. For samples with G16P, 500 µM ligand was added. Experiments were carried out in triplicates. Each time, the samples with and without ligand were prepared with exactly the same procedure from the same protein purification batch. Ellipticity was converted to mean residue ellipticity [θ]. The thermal denaturation data were fitted to the equations describing two-state unfolding by temperature [50,51].

### 2.8. Structure Analysis

Crystal contacts were analyzed using PISA [52]. Root mean square deviations (RMSDs) for structure comparisons were calculated using PDBeFold [53]. All structure figures were generated with PyMOL 2.0 [54]. Interaction figures for ligands were prepared using LigPlot+ [55].

### 2.9. Multiple Sequence Alignment

A multiple sequence alignment of sequences of PGM1 and PGM5 from fishes and mammals (UniProt accession numbers in Appendix A) was generated using Clustal Omega [56]. Appendix A was generated using ESpript 3.0 [57]. Logos were generated using WebLogos [58].

## 3. Results

### 3.1. Genetic Analysis of PGM5 in the Atlantic Herring

Atlantic herring habitats cover a broad range both geographically and in terms of salinity (Figure 2a). Previous whole genome sequencing of the Atlantic and Baltic herring has revealed hundreds of loci associated with ecological adaptation to the Baltic Sea [4]. The *PGM5* locus is located within one of these genomic regions on chromosome 18 showing striking (>80%) delta allele frequency (DAF), the absolute value of allele frequency in the Baltic herring minus the allele frequency in the Atlantic herring (Figure 2b–d, Appendix A). The missense mutation polymorphism (designated rs5164711 in Figure 2e) resulting in either Ala or Val at residue 330 is one out of several single nucleotide polymorphisms (SNPs) in this region showing very high DAF. The Pacific herring has the allele coding for Ala330 (Figure 2e) strongly suggesting that Val330, most common in the Baltic herring, is the derived mutation. The most strongly differentiated SNP in the region is rs5199622 (Figure 2e) for which Atlantic herring populations tend to be fixed for one allele, whereas many of the Baltic herring samples are fixed for the opposite allele. This SNP is located about 35 kb downstream from the *PGM5* Ala330Val SNP and in an intron of a gene annotated as arfaptin-1-like (ENSCHAG00000025243).

A comparison of the allele frequencies for these two SNPs (rs5199622 and the Ala330Val mutation) and salinity reveals very strong, but not perfect, correlations with salinity. The reason for the deviation from a perfect correlation is that the autumn-spawning Baltic herring (population samples 4, 5, 11, 19, and 22) shows a rather low frequency of the Val330 allele compared with the spring-spawning Baltic herring that consistently have a high frequency of this allele (Figure 2e). The results implicate that this genetic adaptation is not directly related to salinity and primarily affects ecological adaptation in the spring-spawning Baltic herring.

In the recently released, chromosome-level genome assembly of the Atlantic herring (GCA_900700415; [4]), *PGM5* occupies a region spanning from 5.16 to 5.18 Mb on chromosome 18. Figure 3 shows the extent of the gene model, as well as the exon–intron structure of the transcripts predicted in the ENSEMBL-provided annotation of the herring genome (database version 99.202). It is important to note that these are predicted transcripts (see Appendix A), based on available RNA-seq data, and not necessarily an exhaustive list of possible transcripts. The exon containing Ala330Val is present in all predicted transcripts. The sequences used for this study (Appendix A) were based on their similarity to other *PGM* genes selected among the expanded list of possible transcripts.

### 3.2. Structure Determination of PGM5 from Herring

The Baltic (bPGM5) and Atlantic (aPGM5) herring PGM5 variants were both expressed in *E. coli* with an N-terminal cleavable His6-tag. Both proteins purified as monomers on size exclusion chromatography and were crystallized in apo form. Crystals of His6-tagged aPGM5 belong to space group P2_1_, while crystals of His6-tagged bPGM5 belong to space group P2_1_2_1_2, both with one molecule in the asymmetric unit. The structures were solved using molecular replacement and refined to 2.25 and 2.05 Å resolution, respectively. Data collection and refinement statistics are found in Table 1. The aPGM5 structure consists of residues 4–567, except for residues 511–513. The bPGM5 structure consists of residues 1–567, except for residues 510–513, as well as four ordered residues from the His6-tag. The active site Ser121 is phosphorylated in the bPGM5 structure in contrast to the structures of aPGM5 and hsPGM1 in apo state [10,12].

Several early studies identified that a divalent metal ion was required for PGM activity, and that the highest activity in most cases was observed with magnesium [59,60,61,62]. Despite crystallization of aPGM5 in a condition without divalent ions, a metal ion is observed in the active site, coordinated at around 2.4 Å distance by Asp292, Asp294, and Asp296 in loop 2. Due to the long coordination distances, the ion was modelled as calcium rather than magnesium (Appendix A). In bPGM5 that was crystallized in the presence of 120 mM calcium acetate, the metal in the active site has 2.0–2.8 Å coordination distances and shows too strong electron density for both magnesium and calcium. Thus, the ion was modelled as Ni^2+^, which because of IMAC purification is the most plausible ion that fits with the coordination distances and electron density (Appendix A). Ni^2+^ has previously been observed in the active site of rabbit muscle PGM1 (PDB ID 1VKL [63]) and can also function in the active enzyme [62]. However, we cannot exclude that another metal ion was picked up during protein expression or purification.

#### 3.2.1. Overall Structure of PGM5

PGM5 shows a bi-lobed structure with four domains arranged around a common active site, forming a deep cleft between domains I–II (residues 1–195 and 196–308) and domains III–IV (residues 309–425 and 426–567) (Figure 4a). As expected, all four domains contribute to the active site with one loop per domain (Figure 4b); loop 1 (residues 119–126), loop 2 (residues 292–297), loop 3 (residues 379–384), and loop 4 (residues 507–520). The proposed catalytic base, Lys393, is located in another loop of domain III [12], here called loop 3B, and the proposed catalytic acid arginine, Arg27, in a loop from domain I [12], here called loop 0. The alternative catalytic acid histidine [17] is an arginine in herring PGM5 and will be further discussed below. In addition, loop 0 (residues 21–27) from domain I lines the active site. Loop 0 and loop 4 upon ligand binding together close the active site.

The structures of apo aPGM5 and bPGM5 are very similar (root mean square deviation (RMSD) of 0.64 Å over 545 C_α_ atoms), but aPGM5 is further closed, showing a shift of up to 2.4 Å of domain IV towards domain I (Figure 4a). When compared with other known structures (Appendix A), the top hit for aPGM5 and second best hit for bPGM5 is the structure of parafusin, a PGM homologue from the unicellular ciliate *Paramecium tetraurelia* (PDB 1KFI, RMSD of 1.2 Å over 522 or 531 C_α_ atoms) [64]. This sulfate-bound structure shows a closed conformation of domain IV compared to the apo structure of the same protein (PDB 1KFQ), which makes it align better with the PGM5 structures (Figure 4c). The movement of domain IV has previously been described as functionally important for the α-D-phosphohexomutases [20,65,66]. For bPGM5, the top hit in structure comparison is to human PGM1 isoform 2 (hsPGM1-2, Appendix A) [10]. In this isoform, the first 77 amino acids are replaced by a new 95 amino acid N-terminus, however with loop 0 conserved in sequence to hsPGM1, with the same fold but an extra helix at the N-terminus. Overall, the structures are also very similar to the structures of human PGM1 (hsPGM1, sequence identity 68 and 70% to isoform 1 and 2, respectively [10,12]), and rabbit PGM1 (ocPGM1, sequence identity 68% [18]) (Figure 4c, Appendix A).

The active sites are very similar in aPGM5 and bPGM5 (Figure 4b) and compared to hsPGM1 (PDB 5EPC), further described below. Loop 4 is disordered in all three structures (Figure 4c). Two structures of ocPGM1 in complex with ligands (PDB 1C4G and 1C47, unpublished) are also highly similar to aPGM5 and bPGM5, but both show severe errors in placement of domain IV in the electron density (also noticed by [19]) and will not be considered further.

#### 3.2.2. Comparison of Crystal Packing in aPGM5 and bPGM5

The only difference between aPGM5 and bPGM5 is the point mutation Ala330Val, that had a clear influence on crystallization. The bPGM5 protein reproducibly crystallized as block-shaped crystals in space group P2_1_2_1_2, where part of the His6-tag is ordered. In contrast, aPGM5 did not crystallize in the same conditions but at other conditions formed small and irreproducible needle crystals in space group P2_1_.

Interestingly, Val330 is involved in a crystal contact in bPGM5 (Figure 5a), but the corresponding interface is not formed in the structure of aPGM5 (Figure 5b). An additional crystal contact is made between the N-terminus and the ordered part of the His6-tag (Figure 5c), while in aPGM5 the N-terminus is involved in a less extensive crystal contact (Figure 5d). Thus, crystal packing shows that Val330 contributes to a protein–protein interaction surface in crystals of bPGM5 that is not observed with Ala330 in the crystals of aPGM5.

### 3.3. Structure of bPGM5 in Complex with Glucose-1-Phosphate

There are no previous data showing PGM activity of PGM5. However, the high similarity of these first PGM5 structures to the active site of PGM1 suggested to us that herring PGM5 may still bind to the corresponding substrate and display enzymatic activity. To this end, crystals of bPGM5 were soaked with the PGM reaction intermediate G16P, and the structure was solved at 1.95 Å resolution. The structure contains all 567 residues plus five residues from the N-terminal His6-tag. Data and refinement statistics are found in Table 1. A close-to-identical structure to slightly lower resolution was solved from a co-crystallization experiment (overall RMSD 0.18 Å over all 572 Cα atoms, data not shown). The complex structure closely resembles apo bPGM5 (RMSD 0.267 Å over 567 Cα atoms, Appendix A) and represents a closed conformation, where loops 0 and 4 form van der Waals interactions to fully close the active site.

Difference density for a ligand is clearly visible in the structure (Figure 6a). Based on the location of the highest peaks in electron density, we conclude that bPGM5 as in the apo structure has a phosphorylated Ser121 in the active site and that the observed ligand is G1P instead of the G16P that was used in crystallization (Figure 6b, Appendix A). In agreement with previous observations for hsPGM1, loop 4 (residue 507–520) orders upon ligand binding [19]. Residues Arg507, Ser509, Thr511, and Arg520 from loop 4 binds the phosphate group of the ligand in a network of hydrogen bonds and forms a lid over the active site, interacting by van der Waals forces to Thr23 of loop 0. The O3 and O4 of the glucose ring is bound by Glu380 and Ser382 of loop 3. Apart from the active-site loops, the only residue interacting with G1P is Lys393, the proposed general base. Lys393 is hydrogen-bonding to O6 of G1P, and both Lys393 and Arg27, the proposed general acid, are within 3 Å of the phosphate group of the activated serine. The active-site metal is modelled as Ni^2+^ and has similar characteristics as in apo bPGM5 (Appendix A).

The complex structure of bPGM5 is similar to the structure of human PGM1 in complex with G6P (PDB code 6BJ0 [19], Appendix A), but domain IV is in a more closed conformation (Figure 6c). G1P and G6P phosphate binding to loop 4 is very similar in PGM1 compared to PGM5, even as the entire domain 4 has shifted closer to domain 1 by up to 5 Å for bPGM5, with loop 4 being around 3 Å closer (Figure 6d). Some differences are observed in the interactions between the ligand and the other loops (Figure 6d). Glu376 in PGM1 is only binding to O3 of G6P, while Glu380 in PGM5 binds to both O3 and O4 of G1P. Lys389 of PGM1 is not directly binding to G6P as compared to the direct hydrogen bond by Lys393 to G1P in PGM5. Arg297 in loop 2 of PGM5 is hydrogen bonding to O6 of G1P, but Arg293 in PGM1 does not seem to bind directly to G6P. In the structure of PGM5, Ser121 is phosphorylated and binding to O6 of G1P, while in PGM1, Ser117 is not phosphorylated and instead binds to O2 by a bridging water. In PGM1, Ser20 of loop 0 is hydrogen bonding to O1 of G6P, an interaction not seen in PGM5, where this residue is an asparagine (Figure 6d).

The same ligand, G1P, is present in a recent structure of human PGM1 isoform 2 (hsPGM1-2, PDB code 6SNO [10], Figure 6c, Appendix A). The active site and the interactions with the ligand are practically identical to the bPGM5 complex structure, despite loop 4 being 2 Å closer to loop 0 in bPGM5 than in hsPGM1-2 (Figure 6e). The main exception is Lys393 in bPGM5 that forms a hydrogen bond to G1P, while in hsPGM1-2, the corresponding Lys407 points away from G1P (Figure 6e). The distance between the phosphorous atom of the phosphorylated serine and O6 of G1P is 3.0 Å in bPGM5 and 3.4 Å in hsPGM1-2 [10]. Since the ligand-bound structures of hsPGM1-2 and bPGM5 show very similar closed states, and the same conformation is observed in structures from soaking and co-crystallization of bPGM5 with G16P (both resulting in bound G1P), this conformation is likely to represent a native state, and we do not expect a more extensive conformational change for catalysis, as recently proposed [10].

### 3.4. Glucose-1,6-Bisphosphate Stabilizes both aPGM5 and bPGM5

As reproducing aPGM5 crystals of sufficient quality for soaking experiments failed, we could only obtain a complex structure with bPGM5. To test whether the Ala330Val mutation affects the ligand binding or thermal stability of PGM5, both PGM5 variants were subjected to thermal denaturation experiments using circular dichroism (CD) and differential scanning fluorimetry (DSF).

Consistent with the crystal structures, both proteins reproducibly displayed CD spectra showing a mixture of α-helix and β-sheet, and no difference in secondary structure content was observed for the two variants (Appendix A). In addition, no change in secondary structure content was detected for either protein by introduction of the ligand, G16P. For all thermal denaturation experiments, the absolute signal at 222 nm decreased upon temperature increase, indicating loss of helicity associated with thermal unfolding of the protein. For both aPGM5 and bPGM5, an increase in melting temperature in the presence of G16P was observed, suggesting the ligand can stabilize both variants (Appendix A). Minor protein precipitation due to an increase in temperature was observed for both variants, and it was very sensitive to initial conditions. Depending on protein batches, freshness of ligand, and sample preparation procedures, the increase in melting temperature (T_m_) due to addition of the ligand varied from 1 to 4 °C for the triplicates in CD-monitored denaturation experiments. No difference in the effect of ligand on the thermal stability was observed between the two variants.

The effect of G16P on the thermal stability of PGM5 was also tested using differential scanning fluorimetry (DSF). In absence of ligand, T_m_ was determined to 61.0 ± 0.1 and 60.7 ± 0.1 °C for tag-free aPGM5 and bPGM5, respectively. In presence of G16P, T_m_ increased to 63.2 ± 0.1 and 62.7 ± 0.1 °C, an increase by 2.2 °C for aPGM5 and 2.0 °C for bPGM5 (Appendix A). Thus, both variants show thermal stabilization upon G16P binding using CD and DSF, but there is no indication that the Ala330Val mutation affects the thermal stability of PGM5 or its thermal stabilization by the ligand.

### 3.5. Enzymatic Activity of PGM5 Is Negligible

Based on the structural evidence of substrate binding and phosphorylation activity, both PGM5 variants were tested for phosphoglucomutase activity using a commercial continuous coupled assay and compared to a positive control, ocPGM1 (Appendix A). For all proteins, either Mg^2+^, G16P, or both were added in order to activate the protein. As the proteins were from different batches than for crystallization, the identity of the bound divalent metal ion is not known, but both ocPGM1 and hsPGM1 display detectable activity with several metal ions including Ni^2+^, however Mg^2+^ provides the highest activity [62,67]. Under all conditions tested, both aPGM5 and bPGM5 displayed phosphoglucomutase activity at least 2000-fold lower than the positive control, barely above the non-catalyzed reaction. No significant difference was observed between aPGM5 and bPGM5. In the positive control ocPGM1, no significant change in activity could be seen upon addition of Mg^2+^, G1P, or both. We conclude that, in agreement with human PGM5 [24,25], herring PGM5 is not an active phosphoglucomutase under the tested conditions. We cannot exclude that the very low activity observed is caused by contamination from the *E. coli* extract.

### 3.6. Sequence Analysis Shows the Main Difference between PGM5 and PGM1 Is in Loop 4

To further map differences between PGM5 and PGM1 that could explain the observed lack of PGM activity, a multiple sequence alignment of PGM5 and PGM1 from a diverse group of 10 fishes and 10 mammals was made (Appendix A). The level of sequence identity to herring PGM5 varies between 63 and 87%. Interestingly, compared to mammalian PGM5, some of the sequences of fish PGM5 show higher sequence identity to human PGM1. As previously shown, PGM1 and PGM5 display high conservation throughout their sequences [8]. Most non-conserved residues are shared in groups either depending on species origin (mammals or fishes) or dependent on protein (PGM1 or PGM5). The active-site loops in general show high conservation between PGM1 and PGM5 (Figure 7). In line with the higher overall similarity, only two consistent differences are observed between fish PGM5 and mammal PGM1: a Ser to Asn substitution in loop 0 (Asn24 in herring PGM5) and a Gly insertion in loop 4 (Gly514 in herring PGM5).

Loop 0, responsible for closing a lid over the active site together with loop 4, is fully conserved in PGM1. The proposed general acid Arg23 in hsPGM1 [14,15,16] is conserved in both proteins. The differences are that fish PGM5 show an Asn substitution of Ser20 in human PGM1, while mammal PGM5 have an Ala/Thr insertion and both Thr19 and Ser20 are replaced by glycines (Figure 7). Since the Thr19Ala mutation in human PGM1 has been shown to cause loss of function [68], this could be one explanation for the lack of phosphoglucomutase activity in mammalian PGM5.

Loop 1 includes the active-site phosphorylated serine, and the recently proposed alternative catalytic acid His118 that forms a hydrogen bond to Ser117 in hsPGM1-1 and hsPGM1-2 (Figure 8a,b) [17]. In PGM5 from some fish, the loop sequence is identical to that in PGM1 (Figure 7). However, PGM5 from herring and spotted gar contain an arginine in place of the histidine (Figure 7, Appendix A). In the PGM5 structures, this arginine is too far away to form an interaction to the corresponding serine. Instead sodium ions bind close to the phosphoryl group to stabilize the negative charge (Figure 8a,b). As the histidine was conserved in all other aligned sequences, also in previous analysis of other sequences [8], a BLAST search for 20,000 similar sequences was conducted [69]. Among these, only the two sequences of PGM5 from fish contain an arginine in this position. The most common motif was TASH (6098), SASH (6065), and TPSH (6822). The hydrogen bond between Thr115 and the phosphoserine in hsPGM1 is clearly important, since mutation to alanine leads to 50% reduction in activity [68], and we found no sequences without either serine or threonine in this position. The replacement of histidine with arginine would possibly explain the lack of activity in herring PMG5, but in another family member, *Pseudomonas aeruginosa* phosphomannomutase/phosphoglucomutase (PMM/PGM), mutation of the histidine in the same position did not lead to complete loss of enzymatic activity [70]. To our knowledge, no mutational study of the TASH-motif histidine has been performed in human PGM1.

The metal-binding loop 2 is conserved in the aligned sequences with the exception of the first glycine that is instead alanine in all except one PGM5 sequence (Figure 7). Loop structures and backbone dihedral angles are very similar for hsPGM1 and herring PGM5, showing that this mutation does not influence the fold of this loop (Figure 8c). This argues against the previous proposal that alanine in loop 2 is one of the reasons for lost activity in PGM5 [25]. PGM2 displays metal-dependent PGM activity [67] despite substitution of the first glycine by proline and the second glycine by alanine [8], further supporting that the change from glycine to alanine in PGM5 is unlikely to affect the functionality of loop 2.

Loop 3 is important for substrate specificity [11,71,72] and strictly conserved in all aligned sequences of PGM1 and PGM5 (Figure 7, Appendix A). Glu376 and Ser378 in hsPGM1 bind to O3 and O4 in the equatorial conformation on G1P and G6P, as needed for PGM activity [11,71,72]. Loop 3 is also close in space to Lys389, the suggested general base that could be sensitive to perturbations in the surrounding environment. Missense lysine-coding mutations of both Glu377 (in loop 3) and Glu388, next to Lys389 in loop 3B (Figure 7) in human PGM1, give rise to insoluble protein, likely by introducing additional positive charge in the active site that can repel the positive charge of the phosphate-binding residues [68,73]. Glu388 is generally conserved in both PGM1 and PGM5 but shows a conservative substitution to aspartic acid in PGM5 from zebrafish and common carp (Figure 7).

The proposed general base of the reaction Lys389 in hsPGM1 [14,15,16] is conserved in all PGM1 and PGM5 and located in a highly conserved region of the sequences (Figure 7).

The phosphate-binding loop 4 is the active-site loop that shows the lowest conservation in fish PGM5, where it is also one amino acid longer than in the other PGMs (Figure 7). This suggests that there is for some reason less selection for conservation of loop 4 in PGM5 from fish. The dynamic behavior of this loop has been shown to be important for PGM1 activity [19]. In hsPGM1, Arg503, Ser505, Arg515, and the backbone amide of Gly506 are forming hydrogen bonds to the phosphate group of G6P (PDB 6BJ0). In bPGM5, the equivalent interactions are formed with G1P, and additional hydrogen bonds are formed with Thr511 and Thr23 (Thr507 and Thr19 in human PGM1) (Figure 6d). Loop 4 is the part of the active site that shows the largest differences between PGM1 and PGM5 [8], but there are also differences between fishes and mammals for both proteins (Figure 7, Appendix A); for example, mammal PGM5 contains four consecutive serines. The observed differences could be responsible for the absent PGM activity of PGM5. Compared to mammal PGM5, fish PGM5 show fewer differences to PGM1 (Figure 7), but all sequences except the one from spotted gar contain a glycine insertion (Figure 7, Appendix A). The conformation of loop 4 is not affected by the glycine insertion close to the phosphate-binding site (Figure 6d,e), but only close to the solvent-exposed tip of the loop (Figure 6c). More likely, the important dynamic behavior of this loop [19] could be affected. Several residues in this region are known mutation sites, and the Arg503Gln and Arg515Leu mutations in human PGM1 lead to complete loss of activity [19].

## 4. Discussion

### 4.1. Why Does PGM5 Lack PGM Activity?

The data presented here show that the structure and substrate binding of herring PGM5 are very similar to PGM1, yet we cannot detect any PGM activity. One early reasoning for the absence of activity in human PGM5 was the mutation of one glycine to an alanine in loop 2 and the mutation of an asparagine to a cysteine in loop 1 [25]. Yet, loop 4 displays the largest differences in sequence between human PGM1 and PGM5 [8], and a recent study points out that the dynamics of this loop are important for functional PGM1 [19]. In addition, mammalian PGM5 has a Thr19Ala substitution in loop 0 (Figure 7) that in human PGM1 kills the activity [68]. Neither of these differences are as pronounced between PGM1 and PGM5 from fish, and in the absence of selection for enzymatic activity, several differences may together explain the lacking activity of PGM5.

The PGM reaction requires the binding of substrate to a phosphorylated enzyme in the presence of a divalent metal ion [62,67]. Based on the PGM5 structures presented here, the protein appears functional in these steps. For bPGM5, we observe the binding of G1P and a metal ion to a phosphorylated enzyme. In aPGM5, Ser121 is unphosphorylated, suggesting that the phosphorylation state is batch-dependent. Despite the different metal ions observed, the structure of the metal-binding site is conserved between hsPGM1 and herring PGM5 (Figure 8c). Previously, Ni^2+^ was observed to activate ocPGM1 and hsPGM1, while Ca^2+^ did not activate hsPGM1, and only gave 0.5% activity for ocPGM1 [62,67]. Despite soaking crystals with the reaction intermediate G16P, G1P is observed in the structure. Most likely, contaminating G1P has bound to the phosphorylated enzyme, but due to the use of different protein batches, we cannot exclude that dephosphorylated enzyme was phosphorylated by G16P to yield G1P. Residues in loops 0–4 show close-to-identical interactions with G1P as in hsPGM1-2 (Figure 6e). One minor difference to hsPGM1-1 bound to G6P is that PGM5 due to the substitution of serine with asparagine lacks the interaction between G6P and Ser20 (Figure 6d).

Next, phosphoryl transfer from enzyme to substrate occurs, creating G16P, with the assistance of the catalytic acid and base (Arg27 and Lys393 in bPGM5). Both catalytic residues are conserved between the active PGM1 and the inactive PGM5 (Figure 7). However, the alternative catalytic acid His118 [17] is substituted by arginine in herring PGM5 (Figure 7 and Appendix A). This replacement might contribute to the absent activity specifically in herring PGM5 (Appendix A). However, in *P. aeruginosa* PMM/PGM, mutation of the corresponding histidine did not completely kill the activity [70], and the substitution is extremely rare even in fish, failing to explain the general lack of activity in PGM5.

Next, G16P has to flip 180 degrees in the active site. This reorientation has been shown to occur on the enzyme without dissociation for both ocPGM1 and *P. aeruginosa* PMM/PGM [74,75] and require the equatorial orientation of O3 and O4 in glucose [71]. At this stage, dynamics are important, involving an unphosphorylated active-site serine [65,66,76], a flexible loop 4, and the interface between domain 4 and the rest of the enzyme [10,19]. Loop 4 in both PGM1 and PGM5 contains several Gly and Ser and thus is predicted to be highly flexible (Figure 7), but PGM5 from fish and mammals show distinct differences that could lead to malfunction in the required loop dynamics.

Lastly, another round of phosphoryl transfer would occur, this time from the intermediate G16P to Ser121 in PGM5. This involves the catalytic acid and base from the first step that perform the same actions in reverse.

In conclusion, based on comparison of structures and sequences of PGM5 and PGM1, we consider most likely that herring PGM5 is defective in the catalytic step or in the reorientation of the intermediate G16P. However, none of our analyses offer a convincing explanation for the lack of enzymatic activity in the entire PGM5 family, and additional experiments will be needed to test the effect of, e.g., mutations in loop 4 on an active enzyme.

### 4.2. Does PGM5 Have the Same Function in Fish as in Humans?

The gnomAD database (https://gnomad.broadinstitute.org) summarizes human mutations found after exome and genome sequencing [77]. This database shows that loss-of-function (LoF) mutations in *PGM5* are significantly underrepresented, suggesting strong negative effects on fitness. The *PGM1* gene, although much more studied than *PGM5*, are more tolerant to the presence of LoF mutations.

PGM1 shows a similar expression in fish as in human, expression is detected in all tissues and is very strong in muscle. In contrast, the expression pattern of PGM5 in different tissues of fish is slightly different from that in human. In herring [1] and zebra fish (https://bgee.org/?page=gene&gene_id=ENSDARG00000060745), RNAseq data indicate the highest expression level in spleen, followed by muscle tissue and swim bladder, while human expression is highest in the bladder and digestive system (https://www.gtexportal.org/home/gene/PGM5, human protein atlas). In human, PGM5 expression is modulated by cell proliferation, with higher levels found in quiescent cells than in growing cells [78], and angiotensin II treatment of quiescent cells leads to increased PGM5 expression [78]. It remains unknown whether this also holds true in herring, but angiotensin II has been linked to osmoregulatory functions in zebrafish [79] and teleost fish [80].

All data regarding human PGM5 indicate that it has a structural role [23,24,25,26,27,28,81], and several protein interactions partners have been identified. The identified interaction partners of human PGM5, dystrophin, utrophin, filamin C, and Xin are also present in the genomes of herring and zebrafish, suggesting that PGM5 can play the same role in these fish. While dystrophin, filamin C and Xin is mainly found in human muscle, utrophin is found in all tissues (http://www.proteinatlas.org) [82]. PGM homologs without PGM enzymatic activity are also present in other groups of organisms, such as the signaling scaffold protein parafusin in the unicellular ciliate *Paramecium tetraurelia* [83].

The present data thus suggest that the presumed structural role of PGM5 is very important, which is also supported by the severe phenotype for PGM5 knockout in zebrafish [28]. It remains to be clarified if the downregulation of PGM5 in cancer is a cause or a consequence [30,31,32,33]. Another example of a conservative mutation in human PGM5 that has a dramatic effect is the somatic oncogenic mutation Ile98Val associated with stomach cancer [34,35]. This isoleucine residue is located in a partly buried interface between domain I and II and is conserved in PGM1 and PGM5 from mammals. Intriguingly, PGM5 from fish sometimes has leucine in this position, but which is equally hydrophobic (Appendix A).

In addition, the functional importance of both PGM1 and PGM5 is supported by the conservation of both proteins since the last common ancestor of “bony vertebrates” [10].

Since most of the active site remains conserved between PGM5 and PGM1, we can speculate that binding of some ligand to PGM5 may modulate its structure and/or dynamics and play a role for affinity to one or several of the protein binding partners. We also cannot exclude that PGM5 may be catalytically active towards a so far unknown ligand, yet to be identified.

### 4.3. Why Is PGM5 Different in the Atlantic and Baltic Herring?

The missense mutation in *PGM5* resulting in the Ala330Val substitution is one of the most striking allele frequency differences between the Atlantic and Baltic herring. We have previously reported that a majority of these missense mutations are expected to be functionally important based on the strong overrepresentation of missense mutations among the sequence variants that distinguish the Atlantic and Baltic herring [1]. However, this mutation is unlikely to be the major cause for the very strong genetic differentiation between the Atlantic and Baltic herring on chromosome 18, because there are some other sequence variants in this region that show an even stronger differentiation, as illustrated in Figure 2e. Despite this, it is fully possible that PGM5 Ala330Val contributes to genetic adaptation to the ecological conditions in the Baltic Sea, since a genomic region under selection may include multiple linked mutations of functional significance [5].

Our analysis of the allele frequency distribution suggests that this signature of selection is not directly related to salinity, because the association is less pronounced for autumn-spawning Baltic herring that is exposed to the same low salinity as spring-spawning Baltic herring (Figure 2a,e). These two subspecies of herring must, besides timing of reproduction, differ in metabolic regulation, because the autumn-spawning larvae show very low growth during their first six months of life when plankton production is negligible. In contrast, spring-spawning herring starves during the winter but should still be ready to spawn when spring comes. Thus, the selection signal on herring chromosome 18 is most likely indirectly rather than directly related to salinity.

Characterization of aPGM5 and bPGM5 using thermal denaturation experiments with CD and DSF and a PGM activity assay shows no clear difference. The thermal stabilization upon binding of G16P is also similar for the two variants (Appendix A). The structures of aPGM5 and bPGM5 are overall very similar and even the backbone conformations at the mutation site are essentially identical. The site of the Ala330Val mutation that distinguishes bPGM5 from aPGM5 shows high variability in PGM5 from fish (Figure 7, Appendix A) varying between Leu (most common), His, Met, Ala, Arg, and in Baltic Herring Val. The corresponding residue in mammalian PGM5 is methionine or isoleucine and in PGM1 with only one exception methionine. The large physicochemical differences of the sidechain in this position indicate that they lack a common function. Some α-D-phosphohexomutases form dimers [64] or higher quaternary structures [20], but the Ala330Val mutation does not change the monomeric state of PGM5 in vitro. The only clear observed difference linked to the mutation is the behavior of aPGM5 and bPGM5 in crystallization. Val330 contributes to a crystal contact in the bPGM5 structures (Figure 5a,b) that does not form in the aPGM5 structure. Thus, Val330 but not Ala330 forms part of a protein–protein interaction surface in the crystal. Selection for two additional methyl groups (Ala to Val) in a surface-exposed location suggests that the mutation may affect the interaction of PGM5 with one of its interaction partners, but no interaction has so far been assigned to this specific region [28]. Possible interaction partners could in such a case be the known partners dystrophin or uthropin, with unknown binding sites, or other, yet-to-be-discovered protein binding partners.

This study illustrates how challenging it can be to explore mechanisms underlying a strong genotype-phenotype relationship in nature. Firstly, it is challenging to disentangle the effects of multiple closely linked variants to be certain which specific variants are important for the relationship. Secondly, in an adaptive process like this, the effect on protein function is expected to be subtle in contrast to the drastic effects of a loss-of-function mutation causing an inherited disorder. However, the facts that Ala330Val occurs in a highly conserved protein, in the near vicinity of highly conserved residues (Figure 7) and our crystallization data indicate that it influences protein–protein interactions suggest that it is plausible that this mutation is functionally important and have contributed to genetic adaptation. However, further work is required to better understand the function of PGM5 in general and its function in herring in particular before a firm conclusion can be drawn.

## 5. Conclusions

The first structures of the enigmatic structural protein PGM5 from herring show high similarity to the main G1P-G6P phosphoglucomutase PGM1. Herring PGM5 binds substrate similarly to PGM1, but lacks enzymatic activity. A structure and sequence comparison suggests that PGM5 may be unable to perform the required substrate flipping and that this is linked to substitutions in the active-site loops that can affect their dynamic properties. Our results form a base for further elucidation of the function of PGM5.

## Figures and Tables

**Figure 1 biomolecules-10-01631-f001:**
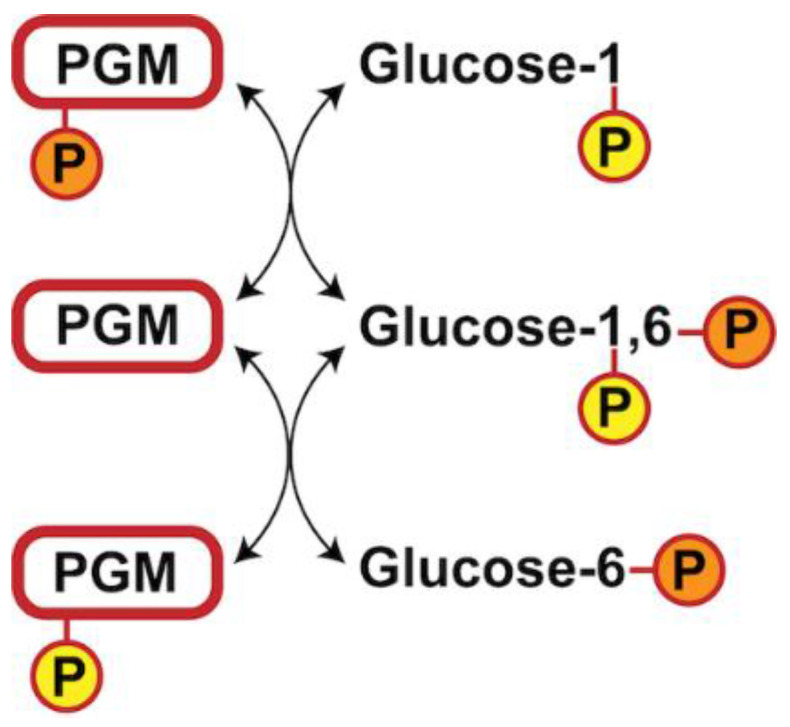
General reaction mechanism of phosphoglucomutase (PGM) enzymes. P = phosphate group.

**Figure 2 biomolecules-10-01631-f002:**
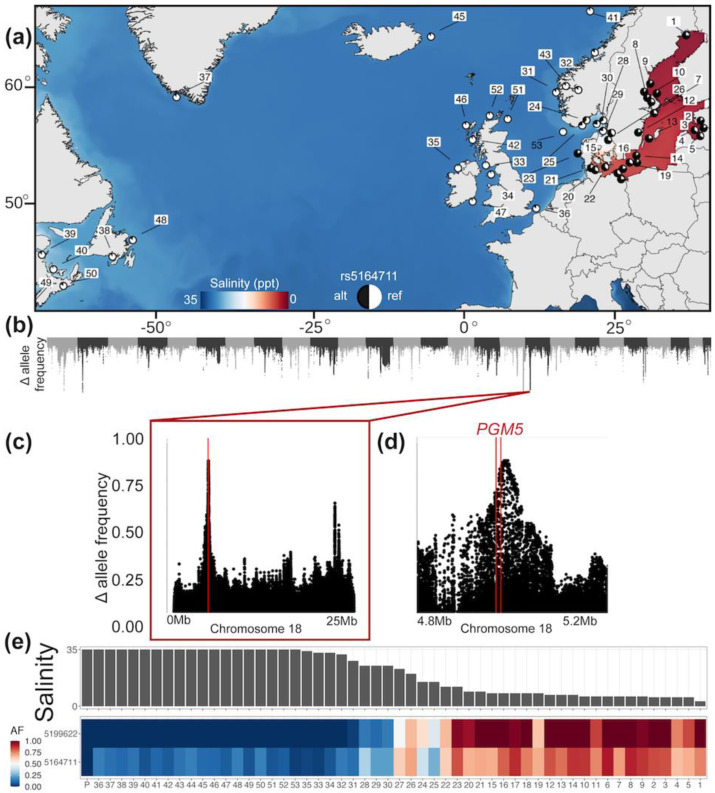
Strong association between variation at the *PGM5* locus in herring and ecological adaptation to the Baltic Sea. (**a**) Geographic distribution of the population samples of the Atlantic and Baltic herring (1–53) analyzed by whole genome sequencing in Han et al. [5]. *X*-axis is longitude and *Y*-axis is latitude. (**b**) Genome-wide screen for allele frequency differences between the Atlantic and Baltic herring. The *P*-value for the most significant single nucleotide polymorphisms (SNP) on chromosome 18 is P = 10^−266^. Pie charts are the frequency of the *PGM5* Ala330Val substitution. (**c**,**d**) Zoom in showing delta allele frequency (frequency in Baltic herring vs. frequency in Atlantic herring); for SNPs on chromosome 18, the location of the *PGM5* gene is indicated. (**e**) Comparison of salinity at each sample location and allele frequencies of the *PGM5* Ala330Val missense mutation (rs5164711) and the SNP (rs5199622) showing the most extreme difference in allele frequency between the Atlantic and Baltic herring. Numbers 1–53 refer to sample locations in Figure 2a, and P represents Pacific herring.

**Figure 3 biomolecules-10-01631-f003:**
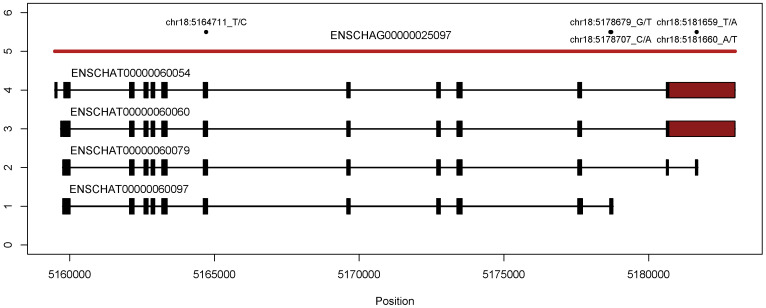
Gene architecture of PGM5 in chromosome 18. The top track (red) shows the genomic footprint of the PGM5 gene model, each subsequent track shows the exon–intron organization of one transcript. SNPs of interest are shown by position and ref/alt base.

**Figure 4 biomolecules-10-01631-f004:**
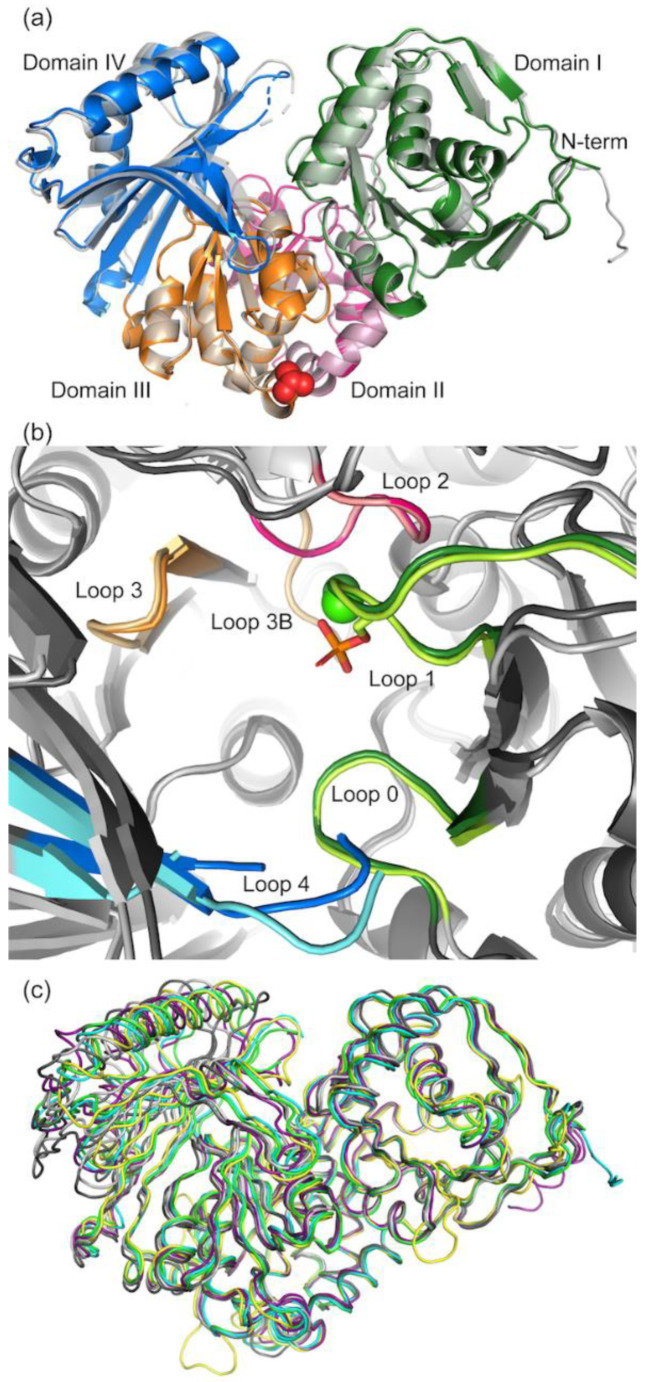
Apo structures of PGM5. (**a**) Overall structure of herring PGM5. aPGM5 is colored according to domain (domain I green, domain II pink, domain III orange, and domain IV blue), while bPGM5 apo is shown in semi-transparent gray. The A330V mutation site is shown as red spheres. The root mean square deviation (RMSD) between the two variants is 0.64 Å over 545 C_α_ atoms. (**b**) Overlay of the active site of apo aPGM5 (dark colors) and bPGM5 (light colors). Active-site loops are colored according to domain as in A. The phosphorylated active-site Ser121 in bPGM5 is shown as sticks and the active-site metal ions are shown as spheres (green—Ca^2+^ in aPGM5, dark green—Ni^2+^ in bPGM5). (**c**) Superposition of PGM structures in ribbon based on domains I-III, for clarity no ligands are shown (aPGM5 apo (green), bPGM5 apo (cyan), human PGM1 (PDB 5EPC chain B, light gray; RMSD 1.67 Å over 529 C_α_ atoms), rabbit PGM1 (PDB 3PMG chain A, dark gray; RMSD 1.58 Å over 481 C_α_ atoms), parafusin with bound sulfate (PDB 1KFI chain A, yellow; RMSD 1.19 Å over C_α_ atoms), and human PGM1-2 (PDB 6SNO chain A, purple; RMSD 1.31 Å over 534 C_α_ atoms)). All RMSD values compared to aPGM5 apo.

**Figure 5 biomolecules-10-01631-f005:**
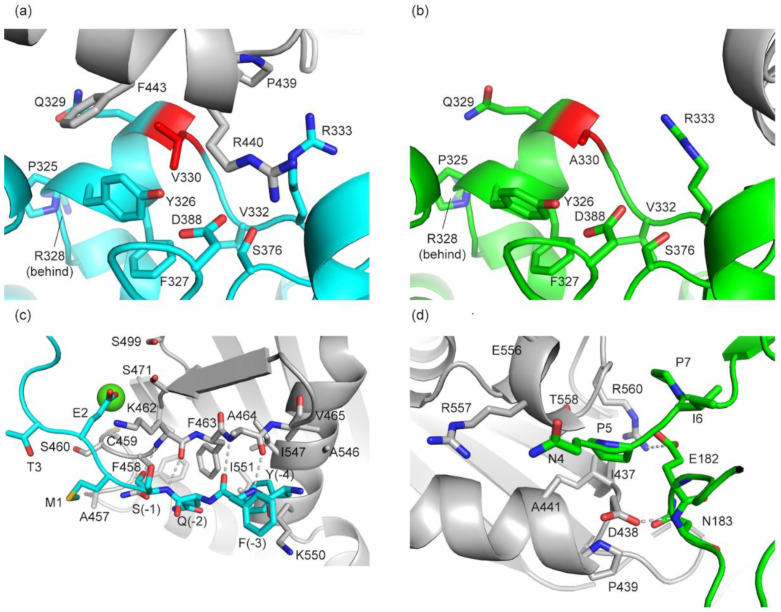
Mutation site and differences in crystal contacts. Symmetry-related molecules are shown in gray. Dashed lines indicate hydrogen bonds. (**a**) Crystal contact of Val330 (red) in bPGM5 (cyan). Side chains within 6Å are shown. (**b**) Corresponding region of Ala330 (red) in aPGM5 (green). Side chains within 6Å are shown. (**c**) β-sheet-extending crystal contact involving 7 ordered residues in the N-terminus of bPGM5 (cyan). A Ca^2+^ ion (green sphere) is observed in this interface, interacting with the symmetry-related molecule, and through bridging waters to Glu2 of the N-terminus. (**d**) Crystal contacts of the N-terminus of aPGM5 (green). Side chains within 6Å are shown.

**Figure 6 biomolecules-10-01631-f006:**
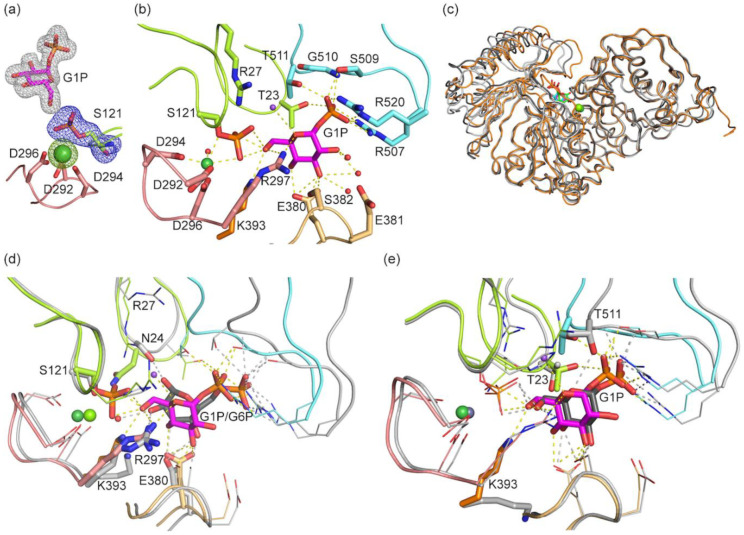
Structure of bPGM5 in complex with G1P. G1P is shown in magenta as sticks, active-site loops are colored according to domain (loops 0 and 1 in green, loop 2 in pink, loop 3 in yellow, loop 3B in orange, and loop 4 in blue). Ni^2+^ is shown as a green sphere. Smaller purple and red spheres represent Na^+^ and waters. Dashed lines indicate hydrogen bonds and metal interactions. (**a**) Polder F_o_-F_c_ omit maps (mesh) of G1P (gray), phosphoserine 121 (blue), and Ni^2+^ (green) contoured at 5σ (0.43–0.45 e-/Å^3^). (**b**) G1P binding site, with G1P, and interacting side chains. (**c**) Comparison of overall structure of bPGM5 in complex with G1P (orange, G1P in magenta), human PGM1 in complex with G6P (6BJ0, chain B, gray, G6P in green), and human PGM1-2 in complex with G1P (6BJ0, chain B, dark gray, G1P in blue). Ni^2+^ (bPGM5, green), Mg^2+^ (hsPGM1, light green), and Zn^2+^ (hsPGM1-2, gray) are shown as spheres. Structures are aligned based on domains 1–3. (**d**) Comparison of the binding sites of G1P in bPGM5 (colors as in b) and of G6P in hsPGM1 (6BJ0, chain B, gray with ligand as sticks and Mg^2+^ in green). Interacting side chains within 6Å of ligands are shown as lines with residues displaying differences highlighted as sticks. (**e**) Comparison of the binding sites of G1P in bPGM5 (colors as in b) and of G1P in hsPGM1 isoform 2 (6SNO, chain A, gray with ligand as sticks and Zn^2+^ in gray). Side chains within 6Å of ligands are shown as lines with residues displaying differences highlighted as sticks.

**Figure 7 biomolecules-10-01631-f007:**
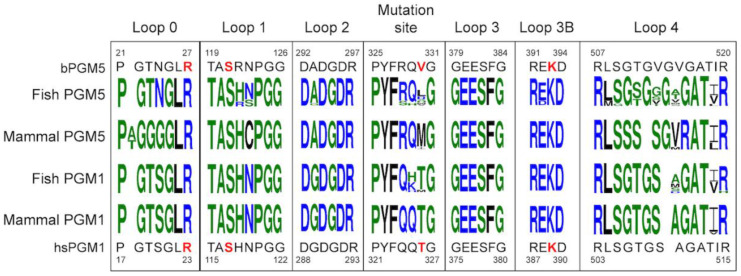
Logo representation of the sequence conservation of active-site loops (loop 0–4) and the Ala330Val mutation region of PGM1 and PGM5 from different species. The logos are based on sequence alignments of PGM5 and PGM1 from 10 fishes and from 10 mammals (Appendix A). The top and bottom sequence and numbering are from the Baltic herring and human, with important catalytic residues (phosphorylated serine, catalytic acid, and base) and the mutation site (Val330 and corresponding amino acid) marked in red and bold. The logos were generated using Weblogos [58].

**Figure 8 biomolecules-10-01631-f008:**
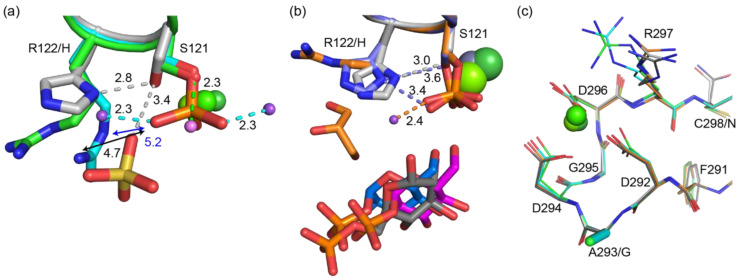
Detailed comparisons of loops 1 and 2 in PGM5 and PGM1. Amino acids are numbered according to herring PGM5. Dashed lines in the same color as the respective structure indicate hydrogen bonds. Mg^2+^ (hPGM1, light green), Ca^2+^ (aPGM5, green), Zn^2+^ (gray, hsPGM1-2), or Ni^2+^ (bPGM5, dark green) ions are shown as spheres. Na^+^ ions in aPGM5 and bPGM5 shown as small light purple or purple spheres. The structures are aligned based on loop1 (**a**,**b**) or loop 2 (**c**). (**a**) Comparison of loop 1 in apo structures of herring PGM5 and hsPGM1. Overlay of aPGM5 apo (green), bPGM5 apo (cyan), and hsPGM1 apo (PDB 5EPC chain B, gray). In bPGM5 apo, Ser121 is phosphorylated. Sulfate bound in hPGM1 shown as sticks. (**b**) Comparison of loop 1 in complex structures of herring PGM5 and hsPGM1. Overlay of bPGM5 (orange) and G1P (magenta) with hsPGM1 (PDB 6BJ0 chain B, gray) and G6P (dark gray) and hsPGM1-2 (PDB 6SNO chain A, blue) and G1P (blue). Sidechains are shown as sticks. In bPGM5 and hsPGM1-2, the serine is phosphorylated. G1P and G6P shown as sticks. (**c**) Comparison of loop 2 in aPGM5 apo (green), bPGM5 apo (cyan), bPGM5 and G1P (orange), human PGM1 apo (PDB 5EPC chain B, light gray), and human PGM1 and G6P (PDB 6BJ0 chain B, dark gray). Sidechains of residues 288–293 in human PGM1 and residues 292–297 in herring PGM5 are shown as lines.

**Table 1 biomolecules-10-01631-t001:** Crystallographic data and refinement statistics.

Structure	aPGM5 apo	bPGM5 apo	bPGM5 + Ligand
PDB code	6Y8X	6Y8Z	6Y8Y
Space Group	P2_1_	P 2 2_1_ 2_1_	P 2 2_1_ 2_1_
Unit cell parameters a, b, c (Å)	59.21, 86.22, 63.52	47.20 94.21 146.69	47.23 94.68 146.97
Unit cell angles α, β, γ (°)	90, 109, 90	90 90 90	90 90 90
Resolution (Å)	55.98–2.25 (2.32–2.25) ^2^	48.90–2.05 (2.11–2.05)	48.99–1.95 (2.00–1.95)
Observations ^1^	72197 (6868)	183748 (9739)	296805 (19464)
Unique reflections ^1^	28235 (2607)	38168 (2184)	48980 (3394)
R_merge_ (I) ^1^	0.232 (0.885)	0.124 (1.023)	0.116 (0.816)
R_meas_ (I) ^1^	0.289 (1.102)	0.140 (1.222)	0.127 (0.898)
R_pim_ (I) ^1^	0.171 (0.648)	0.062 (0.636)	0.051 (0.368)
Mean I/sigma(I) ^1^	3.3 (1.2)	7.5 (2.0)	9.2 (2.0)
CC1/2 ^1^	0.966 (0.221) ^2^	0.948 (0.429)	0.996 (0.454)
Completeness (%) ^1^	98.3(98.5)	91.2 (69.0)	99.9 (99.6)
Multiplicity ^1^	2.6 (2.6)	4.8 (4.5)	6.1 (5.7)
Resolution used in refinement (Å)	55.98–2.25 (2.33–2.25)	44.93–2.05 (2.10–2.05)	47.34–1.95 (1.99–1.95)
No. atoms protein	4337	4423	4482
No. atoms ligands	2	34	63
No atoms waters	216	379	447
R_work_ ^1^	0.2368 (0.3345)	0.1871 (0.4744)	0.1682 (0.2620)
R_free_ ^1^	0.2884 (0.3504)	0.2355 (0.4785)	0.2097 (0.2790)
RMSD bond lengths (Å)	0.002	0.002	0.004
RMSD bond angles (°)	0.473	0.496	0.622
Ramachandran plot (%)			
Favored	95.33	97.50	97.71
Additionally allowed	4.67	2.50	2.29
Wilson B factor (Å^2^)	25.58	26.54	28.60
Protein average B (Å^2^)	35.85	35.31	32.48
Ligands average B (Å^2^)	39.99	49.97	43.01
Water average B (Å^2^)	30.29	39.68	39.87

^1^ Statistics for outer shell is presented in parenthesis. ^2^ Anisotropic diffraction, a resolution cut-off of 2.5 Å gives CC1/2 of 0.417.

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
