# Peer review of "Structure and Characterization of Phosphoglucomutase 5 from Atlantic and Baltic Herring—An Inactive Enzyme with Intact Substrate Binding"

_biomolecules, 2020, doi:10.3390/biom10121631_

Round 1
Reviewer 1 Report
The authors present a detailed study of the protein PGM5 from two species of herring. PGM5 is a largely uncharacterized protein and thus these studies add significantly to our knowledge. A battery of experimental data is presented, including crystal structures, enzyme activity and protein stability assays. Overall the work is of high quality, although there are some issues that may require qualifications in the text (see below). The crystallographic studies provide significant detail regarding the 3D structure of the protein and also its interactions with bound phosphosugars. One of the conclusions of the paper is that PGM5 is highly similar in structure to active, related enzymes, but nevertheless appears to be inactive in this study. This is a bit of a conundrum, but the authors present a detailed analysis of potential differences and suggest a few possibilities. At this point, it is probably not possible to come to a more definitive conclusion. The role of a single residue variant that differs betweeen species, A330V, is established in crystallization but the authors show it appears to have little effect otherwise.
Some of the issues for further consideration are noted here. With regard to the stability assays, did the authors attempt to quantitate the phosphorylation level of the active site serine in their enzymes? This is known to affect thermal stability in this protein family. if not, it should be specified somewhere that this is unknown and may vary batch to batch of protein and change with the addition of phosphosugar ligands. Moreover, the phosphorylation site is sensitive to temperature and may have been changing during the CD and DSF experiments. This should also be noted.
References to the protein superfamily should specify “alpha-D-phosphohexomutase” superfamily, as there are others (beta-phosphohexomutases)
With regard to the lack of enzyme activity, at face value, I agree with the statement that PGM5 does not show significant activity. However, I think that should be qualified – for instance, “under the conditions tested”, or “at equimolar amounts”, etc. There are various factors such as whether all of the protein is bound to Mg2+ that require special consideration to make a more definite conclusion (e.g., treating the protein first with EDTA to remove potentially contaminating metals that reduce activity). The phosphorylation of the enzyme active site serine could also affect activity, and while including G16P in the assay should phosphorylate the protein, that was not demonstrated. Similarly, the oxidation state of the enzymes in this family can affect activity. In the future, a more thorough kinetic study is also warranted, where, for example, higher amounts of PGM5 are tested (10-fold, 100-fold). I don’t expect that additional studies addressing these issues should be included in the manuscript, but there should be some acknowledgment of these factors at appropriate places in the text.
Overall, the paper is well written and the figures are informative.
Reviewer 2 Report
In the manuscript entitle “Structure and characterization of phosphoglucomutase from Atlantic and Baltic herring - an inactive enzyme with intact substrate binding”, the authors characterized Phosphoglucomutase 5 (PGM5) which although closely related to the enzymatically active PGM1, it is inactive as observed with human PGM5, a structural muscle protein. They focus the study on the detected mutation A330V in PGM5, with a very high allele frequency difference between Atlantic and Baltic herring. They have determined the 3D structure of both fish variants (aPGM5 & bPGM5) in their apo forms and the “substrate” bind bPGM5. They also identify intriguing differences between PGM5 from fish and mammals and characterized both systems by CD, DSF, etc.
The article is in general well written and clearly presented. After reading it, I still wonder why in the abstract they focus on the lack of enzymatic activity and do not highlight the structural role of the protein. This happens along the text with, in my opinion, too much background about the un-existent enzymatic activity. I will suggest the authors to strongly reduce the article length by adjusting the relevance of PGM1 to the strict need so that the manuscript gain in concision. It will be more appealing on doing so.
Some minor comments:
- Line 267. Remove “Both structures contain one monomer in the asymmetric unit.” It has been already said after P21212.
- Line 270. Change “The active site serine, Ser121,…” by “The active site Ser121,…”
- Table 1. CC1/2 for the high-resolution shell of 6Y8X is 0.221, un-acceptable low even if I/sigma is still 1.2. Authors should consider reviewing this structure since it looks like 2.5 will be a more reasonable cut-off. It may help to get lower R and Rfree.
- Line 297. What does means that “The crystal packing confirms that PGM5 is a monomer.”? There is a monomer in the ASU but this has already been said.
Reviewer 3 Report
The article ”Structure and characterization of 2 phosphoglucomutase 5 from Atlantic and Baltic 3 herring - an inactive enzyme with intact substrate 4 binding” gives detailed description of PGM5 crystal structures from two types of herrings and investigate the correlations between absence of the phosphoglucomutase activity, emphasizing the major differences in substrate binding loops between active huPGM1 and PGM5. In addition to structural analysis authors use CD and DSF to prove substrate binding in solution and its stabilising effect on both PGM5 variants. Paper is well written with justified conclusions. It will benefit from more focused presentation and illustrations.
Authors mentioned possible binding partners of PGM5 and effect of A330V substitution for altered partners interactions. This part is very speculative and based on difference in crystal packing it doesn’t add value to the manuscript.
The speculation on the importance of the A330V substitution in PGM5 to the adaptation of herrings to the salinity was not confirmed either, so there is no need in too much emphasis on this hypothesis in the introduction either.
Minor comments
- Abstract mentions protein binding partners differences, but authors did not provide enough justification for this.
- Page 2 lines 78-84. It is not clear which protein authors speak about PGM5 or PGM1. Please make it clearer
- Page 4 line 137, is 10 column volumes relates to buffer A? make it clearer
- Page 4 line 159 please add wavelength of data collection, beam line and detector.
- Page 7 figure 3 please change the font in the top panel - it is unreadable at 100% magnification
- Page 7 line 267 remove sentence “Both structures contain one monomer in the asymmetric unit” you already mentioned.
- Page 7 line 268 add “collection” after “data”.
- Page 8 CC1/2 0.222 is too low, please cut the data to CC1/2 0.3. This may improve the poor Rmerge/Rpim of the data
- bPGM5 geometry is too tightened for the resolution, it has to be relaxed to 0.005-0.010. There are no validation reports to access the quality of the structures.
- Page 9 line 320 please include r.m.s.d into the figure caption
- Page12 figure 6, too many hydrogen bonds make figures less appealing, it will be better to leave only important bonding and use different colours for waters belonging to different structures.
